# GTR: GAN-Based Trusted Routing Algorithm for Underwater Wireless Sensor Networks

**DOI:** 10.3390/s24154879

**Published:** 2024-07-27

**Authors:** Bin Wang, Kerong Ben

**Affiliations:** College of Electronic Engineering, Naval University of Engineering, Wuhan 430033, China; benkerong08@163.com

**Keywords:** underwater wireless sensor networks (UWSNs), routing algorithm, trust evaluation model, generative adversarial network (GAN), Q-Learning

## Abstract

The transmission environment of underwater wireless sensor networks is open, and important transmission data can be easily intercepted, interfered with, and tampered with by malicious nodes. Malicious nodes can be mixed in the network and are difficult to distinguish, especially in time-varying underwater environments. To address this issue, this article proposes a GAN-based trusted routing algorithm (GTR). GTR defines the trust feature attributes and trust evaluation matrix of underwater network nodes, constructs the trust evaluation model based on a generative adversarial network (GAN), and achieves malicious node detection by establishing a trust feature profile of a trusted node, which improves the detection performance for malicious nodes in underwater networks under unlabeled and imbalanced training data conditions. GTR combines the trust evaluation algorithm with the adaptive routing algorithm based on Q-Learning to provide an optimal trusted data forwarding route for underwater network applications, improving the security, reliability, and efficiency of data forwarding in underwater networks. GTR relies on the trust feature profile of trusted nodes to distinguish malicious nodes and can adaptively select the forwarding route based on the status of trusted candidate next-hop nodes, which enables GTR to better cope with the changing underwater transmission environment and more accurately detect malicious nodes, especially unknown malicious node intrusions, compared to baseline algorithms. Simulation experiments showed that, compared to baseline algorithms, GTR can provide a better malicious node detection performance and data forwarding performance. Under the condition of 15% malicious nodes and 10% unknown malicious nodes mixed in, the detection rate of malicious nodes by the underwater network configured with GTR increased by 5.4%, the error detection rate decreased by 36.4%, the packet delivery rate increased by 11.0%, the energy tax decreased by 11.4%, and the network throughput increased by 20.4%.

## 1. Introduction

Underwater is an important environment for human survival and development. Underwater wireless sensor networks carry service data for underwater environmental monitoring, channel management, regional prevention, and disaster warnings and transmit important information related to ecological, navigational, facility, and maritime safety [1,2]. Therefore, it is particularly important to achieve secure data transmission in underwater wireless sensor networks.

In underwater wireless sensor networks, packets are broadcast based on acoustic media, and these packets are completely exposed to the open underwater environment. Owing to the influence of water flow, the neighbor relationships between nodes change dynamically, and some nodes must be periodically updated due to energy depletion [3], which exacerbates the dynamic range of node relationships in the network. In this open and dynamic transmission environment, malicious nodes can often infiltrate the network and carry out network attacks, disrupting normal data forwarding. Network attacks can be a combination of one or more methods, and the attack process can also be intermittent [4,5]. To improve the security of packet forwarding in underwater wireless sensor networks, security technologies such as encryption, authentication, intrusion detection, synchronization, and location are widely used in various studies. Trust management is a security technology that can be applied to underwater wireless sensor networks. It analyzes and evaluates the trust of nodes by collecting the location, energy, signal, and packet forwarding features of nodes over a period and updating the trust of nodes based on changes in the underwater transmission environment to avoid malicious nodes participating in the packet forwarding process and causing packet security threats [4,6]. Therefore, selecting trusted nodes as nodes on the transmission route is an important foundation for achieving secure transmission in underwater wireless sensor networks.

The underwater acoustic channel has problems such as a large transmission delay, limited transmission bandwidth, large signal attenuation, multiple interference factors, and severe multi-path phenomena, which also vary with time and space [1]. Problems caused by the poor transmission channels are difficult to distinguish and detect from the packet loss, transmission delay, retransmission, and energy consumption caused by malicious node attacks. To distinguish between malicious nodes and trusted nodes and achieve node trust evaluation, a method or classifier for calculating node trust must be designed to distinguish between trusted nodes and malicious nodes. ARTMM [6] designed a trust evaluation method from multiple perspectives such as transmission links, data, and nodes. TUMRL [7] collects trust evidence from multiple indicators, such as communication, data, and energy consumption, and considers the node’s mobility model and the trust of nodes on the transmission route. SVM [8,9], decision tree [10], DS evidence theory [9], K-means [11], and DBSCAN [12] have been used to construct classifiers for trust evaluation. LSTM is used to train the trust feature models of nodes, whereas reinforcement learning is used to obtain optimal node trust evaluation policies [13]. However, owing to the lack of valuable and labeled training data, there is a serious imbalance between the state data of trusted nodes and the abnormal data of malicious nodes, which makes it difficult to optimize the malicious node classifier [12]. When unknown malicious nodes break into an underwater network, they often lead to a significant loss in the performance of the trust evaluation model.

GANs consist of two parts: a generator and a discriminator. Through mutual confrontation and game, the abilities of the generator and the discriminator are enhanced [14]. After multiple rounds of training, the generator can generate forged data that are difficult to distinguish based on noise, and the discriminator can improve its ability to distinguish between real data and fake data. GANs are widely used in image processing, natural language processing, and anomaly detection [14], and have shown a good performance in industrial IoT anomaly detection and trust evaluation [15,16]. GANs belong to unsupervised models and the training of these models does not require labeled training data. Moreover, GANs come with a generator that can augment training data, solving the problem of insufficient training caused by data scarcity and imbalance [16]. GANs can also be embedded into deep learning models to automatically capture the feature information in training data and improve discrimination ability. Although GANs have higher computational power requirements, compared to the energy consumption of underwater acoustic communication, their computational energy consumption is negligible [17]. With the enhancement of computing chip capabilities, the data processing capabilities of underwater nodes will be increasingly enhanced, and the advantage of computing energy consumption in exchange for communication energy consumption will become more significant. Applying GANs to underwater nodes to achieve trust evaluation and support secure transmission route generation has become a practical approach.

The dynamic and continuous changes in the underwater transmission environment in both space and time cause the selection of the next-hop node in the packet forwarding process to have Markov characteristics. The selection of the next-hop node in underwater wireless sensor networks depends on the historical forwarding experience and current transmission conditions. Reinforcement learning methods are often used to solve Markov problems, and applying reinforcement learning to routing selection in underwater wireless sensor networks can significantly improve the adaptability of the routing selection. In an open underwater environment with malicious nodes, combining the node trust evaluation model with an adaptive routing selection algorithm can enable the transmission route to effectively avoid malicious, isolated, far, dead, and busy nodes, achieving secure, efficient, and reliable packet forwarding for underwater wireless sensor networks.

Therefore, this article aims to improve the comprehensive performance of packet forwarding in underwater wireless sensor networks and proposes a trusted routing algorithm, GTR, based on generative adversarial networks and reinforcement learning algorithms. The main contributions include:(1)We propose a node trust evaluation matrix consisting of multi-period transmission nodes, links, data feature values, and their differential values. The trust evaluation matrix can well characterize the trust features of trusted nodes, and based on the trust evaluation feedback of neighboring nodes, it can maximize the detection rate of malicious nodes and reduce the error detection rate.(2)We propose a GAN-based node trust evaluation model. The evaluation model utilizes a generator to continuously generate forged trust feature profiles of trusted nodes and improve the discriminator’s ability to distinguish the trust feature profiles of trusted nodes through training. When the transmission environment changes and the trust feature profiles of trusted nodes become blurred, a trigger update method is adopted to ensure the stability of the detection ability of the trust evaluation model.(3)We propose an adaptive routing algorithm based on Q-Learning. This adaptive routing algorithm establishes transmission routes between trusted nodes by considering their location, remaining energy, topology relationships, and historical data forwarding. It enables packet forwarding to avoid malicious nodes and adaptively select routes based on changes in the transmission environment, effectively improving the overall performance of the underwater wireless sensor network packet forwarding.(4)Simulation experiments show that the proposed GTR algorithm overcomes the impact of unlabeled and imbalanced training data, and compared to baseline algorithms, GTR can provide a better malicious node detection performance and data forwarding performance. Under the condition of 15% malicious nodes mixed with 10% unknown malicious nodes, it can improve the detection rate of malicious nodes by 5.4%, reduce the error detection rate by 36.4%, increase the packet delivery rate by 11.0%, reduce the energy tax by 11.4%, and increase the network throughput by 20.4%.

The remainder of this article is organized as follows: Section 2 covers related works; Section 3 presents the system model; Section 4 details the trust evaluation algorithm; Section 5 explains the adaptive routing algorithm; Section 6 consists of a performance evaluation; Section 7 is the discussion; and Section 8 is the conclusion.

## 2. Related Works

This article introduces the latest research on secure routing and trust evaluation in underwater wireless sensor networks.

### 2.1. Research on Secure Routing Algorithms

The goal of routing algorithms for underwater wireless sensor networks is to forward packets from underwater sensor nodes through communication nodes hop by hop. To achieve secure packet forwarding, researchers often need to consider the location, remaining energy, and neighboring nodes of underwater nodes when selecting the next-hop node. Table 1 summarizes several typical underwater wireless sensor network routing algorithms and analyzes the methods of route establishment and the design of secure packet forwarding.

VBF proposed by Xie et al. [18] selects the node closest to the target vector as the next-hop node. QELAR proposed by Hu et al. [19] is based on Q-Learning. When selecting the next-hop node, it not only selects the node on the shortest path, but also considers the remaining energy of the node and its energy consumption balance, avoiding the problem of the premature energy depletion of nodes on the optimal path. DQIR proposed by Geng et al. [20] dynamically selects routes based on deep reinforcement learning to balance the energy and distance of neighboring nodes to the target node, further improving the performance of packet forwarding. QTAR proposed by Nandyala et al. [21] considers network topology to determine the candidate forwarding nodes and uses Q-Learning to obtain the optimal decisions for transmission routes. DROR proposed by Wang et al. [22] combines reinforcement learning with opportunistic routing to ensure real-time and efficient packet forwarding, and can adaptively bypass invalid nodes during transmission route construction. SQMCR proposed by Wang et al. [23] adopts a cooperative communication approach, in which, after selecting the next-hop node, the required cooperative nodes are selected based on game theory and reinforcement learning algorithms, further improving the reliability of packet forwarding. By incorporating reinforcement learning into routing algorithms, packets can learn from historical forwarding experience when selecting the next-hop node, thus effectively reducing the blindness of routing selection.

However, when malicious nodes are mixed into a network, nodes with good comprehensive conditions may not be the best choice. In particular, for some underwater critical and sensitive applications, it is important to ensure that the forwarded packet on the transmission route is not blocked, intercepted, tampered with, or replayed and that the confidentiality, integrity, data freshness, and network availability are not damaged. To achieve secure packet forwarding, the lightweight encryption algorithm is applied to data encryption transmission, integrity protection, and node authentication. Chhaganet al. proposed a hybrid security framework [24] that includes physical layer security, software-defined networking, node cooperation, cross-layering, context awareness, and cognition to adapt to environmental changes, network states, and possible attack forms. SEEORVA [25] proposed by Varun et al. uses lightweight encryption technology to encrypt transmitted data. SecFUN [26] proposed by Giuseppe et al. uses AES and a short digital signature algorithm in Galois counter mode as encryption blocks to ensure the confidentiality, integrity, authentication, and non-repudiation of forwarded data. Hu et al. proposed a method for the data collection, transmission, and storage of the marine Internet of Things [27], using the elliptic curve algorithm to ensure the confidentiality, reliability, and integrity of data forwarding and using the consensus algorithm and blacklist mechanism to detect and handle faulty or malicious nodes. RSN^2^ proposed by Prakash et al. [28] used an improved data encryption standard (I-DES) to protect data security and combined the packet forwarding process with encryption and decryption processes to further enhance data forwarding security. To further ensure routing security, methods such as secure location, effective signal coverage, and node authentication have been used. SARP proposed by Manjula [29] uses the direction of the arrival of signals to authenticate the security of neighboring nodes. R-CARP proposed by Angelo et al. [30] uses a short digital signature algorithm for node authentication to meet the requirements for use in underwater-constrained environments. Sabir et al. proposed a secure and lightweight UWSNs key management framework [31] using an elliptic curve algorithm for key distribution and certificate revocation list for key revocation. ST-CJ proposed by Su et al. [32] uses the long propagation delay of underwater acoustic signals to generate interference signals based on cooperation conflicts at eavesdroppers without affecting the reception of legitimate users, which forms a security coverage of the network. The AFSA-ACOA-SC routing algorithm proposed by Wang et al. [33] uses the artificial fish swarm (AFS) and ant colony optimization (ACO) algorithms to construct routes, fully considering the problem of acoustic curve transmission to avoid mixing malicious nodes into the transmission route. To further improve data security, researchers rely on querying and monitoring the status of neighboring nodes to distinguish whether malicious nodes have been mixed into these neighboring nodes. SEECR proposed by Khalid et al. [34] determines whether to classify a neighboring node as a malicious node by monitoring the number of times that packets are leaked from the neighboring node and excludes it from the scope of the next-hop node. Yang et al. proposed a trusted routing algorithm based on blockchain and reinforcement learning [35], which manages the trust of nodes through the blockchain to make their routing information traceable. DOIDS [12] proposed by Zhang et al. for opportunistic routing selects the energy consumption, forwarding, and link quality information of candidate nodes as detection features, and detects potential malicious nodes through the DBSCAN clustering algorithm to reduce the false detection rate.

Although the application of encryption and signature algorithms in the construction of transmission routes can certainly achieve secure data forwarding, achieving secure and reliable key updates in limited underwater networks is exceptionally difficult. Moreover, the need for the encryption and decryption of every packet sent imposes a significant burden on underwater nodes with limited energy. Finding trusted nodes by controlling the signal transmission path is also challenging in time-varying underwater transmission channels. However, by monitoring the forwarding information between neighboring nodes and using trust evaluation methods to find trusted next-hop nodes, this is easier and lighter to implement in underwater networks. Therefore, as an effective and lightweight method to avoid malicious nodes, trust evaluation is combined with a routing algorithm that can improve the security of packet forwarding.

### 2.2. Research on Trust Evaluation Algorithms

Under the assumption that underwater wireless sensor network nodes are homogeneous, the higher the matching degree between the target node’s trust feature and the trusted node’s trust feature, the higher the target node’s trust, and the lower the matching degree, the lower the target node’s trust. Table 2 summarizes several typical underwater wireless sensor network trust evaluation algorithms, focusing on the evaluation indicators, classification method, and update method.

The choice of features to describe the node’s trust is a key issue. ARTMM [6] proposed by Han et al. defines the transmission link, data, and node status as evaluation indicators to describe the node’s trust. The transmission link status includes the utilization rate, packet loss rate, and packet error rate, which represent the link capacity and reliability. The data status includes the data integrity. The node status includes the data forwarding frequency and remaining energy, which represent the node’s honesty and competitiveness. The calculation of node trust also considers the impact of water flow on the node’s location. TMC [36] proposed by Jiang et al. uses cloud theory to construct a trust evaluation model. TUMRL [7] proposed by He et al. uses link, data, and energy status as the evaluation features for node trust, and assigns different trust weights to nodes based on their trust in the transmission route to ensure the safety and reliability of critical nodes. ITrust [37] proposed by Du et al. also incorporates environmental status information into the evaluation indicators for node trust. LTrust [13] proposed by Du et al. even uses the recommendations of neighboring nodes as the basis for evaluating node trust.

Researchers have proposed various methods for calculating the trust of nodes based on their feature attributes, as well as methods for implementing the classification of trusted and malicious nodes. ARTMM [6] uses fuzzy logic and other methods to normalize the evaluation indicators for trust and uses direct calculation methods to obtain the trust of nodes. However, due to the dynamic nature of underwater wireless sensor networks, the weights of trust feature attributes can change with the transmission environment of node deployment. Therefore, training a classification model based on the underwater network state can achieve better results. STMS [8] proposed by Han et al. uses fuzzy logic to obtain the node states for training a classifier based on an SVM to achieve the classification of trusted and malicious nodes. SDFTM [9] proposed by Su et al. combines DS evidence theory with an SVM to construct a classifier. TEUC [10] proposed by Jiang et al. uses the C4.5 decision tree to construct a classifier. To enhance the learning ability of trust features, LTrust [13] uses LSTM to calculate and evaluate node trust. However, due to the special nature of the underwater environment, it is difficult to obtain labeled underwater state information, and researchers have proposed using unsupervised learning algorithms to construct classifiers. TMIS [11] proposed by Liang et al. uses K-Means++ and SVM to construct a classifier. DOIDS [12] uses the DBSCAN clustering algorithm to distinguish malicious nodes. However, malicious nodes generally occupy a minority in underwater networks, and malicious nodes also appropriately control their attack behavior to better hide themselves, which causes an imbalance in the training data. Therefore, node trust evaluation also needs to overcome the impact of data imbalances. ITrust [37] uses the lonely forest algorithm to construct an unsupervised classifier in imbalanced data. In the field of the industrial IoT, Yang et al. proposed a GAN-based trust evaluation model, GALTM [16]. GANs belong to unsupervised classifiers, where the generator can automatically generate data based on noise and continuously input the generated data and real data into the discriminator for training, which can greatly alleviate the problem of insufficient training in the discriminator due to imbalanced data. Therefore, GALTM has achieved a good performance in the application of node trust evaluation in the industrial internet of things. GANs have been widely used in anomaly detection, providing a good research idea for solving the problem of underwater network node trust evaluation.

Due to the dynamic nature of the underwater network environment, the features of trusted nodes change with the transmission environment. This requires the node trust evaluation model to be updated as the transmission environment changes. ARTMM [6] uses a time window method to periodically update the node trust and uses a forgetting factor to reduce the influence of earlier node trust. LTrust [13] uses a recurrent neural network based on LSTM to update the node trust. TUMRL [7] treats the process of updating the node trust as a Markov process and uses reinforcement learning to derive the next state of the node trust. However, the node trust does not change periodically or continuously. Using a periodic method to update the node trust will inevitably result in significant computational power and energy consumption. Both DOIDS and GALTM use a triggered update method, which starts the updating of the evaluation model when the classification accuracy of the evaluation model cannot meet the threshold requirements as the transmission environment changes. During the updating process of the evaluation model, using a fine-tuning method can greatly improve the efficiency and accuracy of the evaluation model update.

Trust evaluation algorithms, whether based on supervised or unsupervised algorithms, face difficulties in achieving convergence in model training when dealing with imbalanced node trust feature data in underwater networks. The time-varying transmission environment of underwater networks often renders the trained trust evaluation model ineffective, while the periodic update method may lead to either overly frequent or overly delayed updates. Trust evaluation algorithms based on GANs primarily focus on the trust features of trusted nodes and trigger updates to the trust evaluation model based on changes in these features, effectively enhancing the detection performance of malicious nodes in underwater networks.

According to the above research, the feature indicators required for node trust evaluation are highly correlated with the evaluation indicators for transmission routing selection. The state information of neighbor node interactions can support both node trust evaluation and routing selection. To overcome the impact of malicious nodes on the packet forwarding security, it is necessary and important to incorporate the trust evaluation results of nodes into the conditions for routing selection when constructing a trusted route for underwater wireless sensor networks. Since the forwarding process of packets follows a Markov process, using reinforcement learning routing algorithms can better improve the performance of packet forwarding. GANs are suitable for constructing classifiers under imbalanced training data conditions, so a trust evaluation model based on a GAN can better meet the evaluation needs of node trust in underwater wireless sensor networks. As the computing power of underwater nodes gradually improves, the energy consumption used for a reasonable computing power will be more efficient than that used for blind communication.

## 3. System Model

During the packet forwarding process of underwater wireless sensor networks, malicious nodes may attack and compromise the security of packet forwarding. Therefore, by identifying malicious nodes and excluding them from the transmission route, the security of packet forwarding can be improved.

### 3.1. Analysis of Network Attack Modes

Underwater wireless sensor networks are in open space. To destroy the confidentiality, integrity, freshness, and availability of data, malicious nodes often use the following common attack modes [4].

(1)Black Hole Attack: Packets are forwarded to a malicious node and then “swallowed” without being forwarded further.(2)Wormhole Attack: Two malicious nodes establish a fast transmission channel to “lure” other nodes to forward data to the malicious nodes.(3)Sink Hole Attack: A malicious node masquerades as a target node and “lures” other nodes to forward data to the malicious node.(4)Sybil Attack: Malicious nodes send multiple lots of false location information to the outside world, “luring” other nodes to forward data to the malicious nodes.(5)Selective Forwarding Attack: Malicious nodes no longer blindly discard packets, but instead selectively forward them. So, it is difficult to distinguish between the packet loss rate caused by malicious node attacks and the packet loss rate caused by poor channels.(6)Exhaustion Attack: Malicious nodes frequently send false packets to the network to exhaust the energy of nodes on the transmission route.(7)Jamming Attack: Malicious nodes disrupt normal node communication by releasing jamming signals.(8)Acknowledgment Spooling Attack: In many cases, the sending node needs to receive an acknowledgment message from the receiving node before it considers the packet to have been successfully received. In the acknowledgment spooling attack, a malicious node sends a spoofed acknowledgment message to create the illusion that the packet has been successfully received.

Various types of network attacks carried out by malicious nodes are either undertaken in a single way or in a combined way. The process of network attacks is also intermittent, and the style of network attacks is constantly changing. If the premise of homogeneous network nodes is taken into account and the impact of the underwater transmission environment is comprehensively considered, the node, channel, and data state of malicious nodes in multiple periods can be used as feature indicators and compared with the relevant feature indicators of trusted nodes one by one to discover the traces of malicious nodes.

### 3.2. Trusted Routing Model

Underwater wireless sensor networks generally include underwater sensor nodes, communication nodes, and surface sink nodes. Data are collected by underwater sensor nodes, transmitted hop-by-hop through communication nodes using underwater acoustic channels, and forwarded to surface sink nodes. Finally, the surface sink nodes transmit the data to shore-based or ship data centers through wireless channels. Combining “trust evaluation” with “routing technology” to construct a trusted routing model allows transmission routes to avoid malicious nodes, providing good protection against network attacks from malicious nodes.

According to the attack methods of malicious nodes, the location, signal activity, signal strength, energy consumption rate, packet loss rate, packet error rate, and other features of nodes can be used as the basis for distinguishing trusted nodes from malicious nodes. The trust evaluation model based on a GAN generates a trusted node feature profile based on the above features. The trusted node feature profile is compared with the feature profiles of other nodes by the GAN-based trust evaluation model. Nodes that exceed the threshold range are determined to be malicious nodes. Within the single-hop coverage range of underwater acoustic signals, neighboring nodes are homogeneous and have similar feature attributes. During data forwarding, the current node where the packet is located is used as a trusted node to perform trust feature portraying and form a feature profile. This feature profile is compared with the feature profiles of neighboring nodes within the signal coverage range of the node by the trust evaluation mode, and malicious nodes are found and excluded from the transmission route.

In underwater wireless sensor networks, neighboring nodes synchronize their states by sending their status to each other. The current node where the packet is located selects the next-hop node based on the location, remaining energy, transmission delay, and historical forwarding of its neighboring nodes. Firstly, the trusted nodes with the shallowest water depth, enough remaining energy, and without malicious neighboring nodes are selected as the trusted candidate next-hop nodes set. Then, based on the reinforcement learning routing algorithm, the optimal route is selected from the trusted candidate next-hop nodes set according to the optimization goal and historical forwarding experience. Through forwarding using the best-trusted nodes one by one, a multi-hop data forwarding trusted route is formed, and finally, the secure forwarding of data is achieved.

When the underwater transmission conditions change, the trust features of the trusted nodes may become blurred or even change, and even the trusted node’s feature attributes cannot fully match the trained trusted node feature profiles. Then, it is necessary to initiate an update of the GAN-based trust evaluation model. During the updating process, it is necessary to re-learn the current node, that is, the feature attributes of the trusted nodes. During the learning process, through the fine-tuning of parameters, a new trusted node feature profile is quickly established, and the trust evaluation model quickly converges. Then, based on the trained evaluation model, the neighboring nodes are re-evaluated.

Figure 1 depicts the trusted routing model of underwater wireless sensor networks with the three procedures of trust evaluation, adaptive data forwarding, and model triggering updates. The transmission route is adaptively adjusted based on the results of a trust evaluation and the routing selection conditions, ensuring that packets are sent by underwater sensor nodes, forwarded hop-by-hop by communication nodes, and securely delivered to the surface sink node.

## 4. Trust Evaluation Algorithm

The trust evaluation algorithm is used to distinguish between trusted nodes and malicious nodes and is the basis for achieving secure data forwarding. The trust evaluation algorithm collects node statuses to form a trust evaluation matrix, builds a trust evaluation model based on a GAN, and improves the ability to distinguish malicious nodes by learning the trust evaluation matrix over multiple periods. The ability of the trust evaluation model to distinguish will deteriorate as the transmission environment changes. When this degree of deterioration exceeds a threshold, the evaluation model is triggered to update. The evaluation model update uses parameter fine-tuning to ensure the detection performance and update efficiency of the evaluation model.

### 4.1. Trust Feature Attribute

The trust feature attribute includes the node trust features, the transmission channel trust features, and the data trust features, as shown in Figure 2. The nodes in the underwater wireless sensor network comprehensively judge the trust of the target node by monitoring the features of the target node itself, the transmission channel with the target node, and the forwarded packets. 

#### 4.1.1. Node Trust Feature

In underwater wireless sensor networks, the main role of communication nodes is underwater data forwarding. The node location is the key to underwater communication, and the node communication ability reflects the state of node participation in communication. Therefore, under the premise of homogeneous node attributes in the network, the node trust feature is reflected in their location feature and communication ability feature.

(1)Node Location Feature Attribute

The node location feature reflects the topological relationship of nodes in the network and the impact of water flow changes. Considering an underwater wireless sensor network with evenly distributed nodes, under the premise of node homogeneity, within the range of neighboring nodes, each node will be affected by water flow to a similar extent, and changes in the network topology of these nodes will also be similar. The location relationship of the nodes in the network topology is reflected in changes in the neighboring nodes and the location vector between the target node and the reference node. Therefore, the change amount of the neighboring node and the change amounts of the distance between the neighboring node and the reference node, the vertical angle, and the horizontal angle are selected as the trust feature attributes of the node location.

The change amount, $\Delta Neigh_{j}$ of the neighboring nodes of the target node $N_{j}$ can be calculated using the number of neighboring nodes $N_{before}^{j}$ in the previous period and the number of neighboring nodes $N_{now}^{j}$ in the current period [38], as shown in Formula (1).
(1)$$\Delta Neigh_{j}=\frac{\left|N_{before}^{j}\cap N_{now}^{j}\right|}{\left|N_{before}^{j}\cup N_{now}^{j}\right|},$$

The locational relationship between the reference node $N_{i}$ (located at point O) and the target node $N_{j}$ (located at point A) is shown in Figure 3. The distance between the two nodes is $Dist_{i,j}=\left|\vec{OA}\right|$, with a vertical angle of α and a horizontal angle of β. The distance change $△{\mathrm{Dist}}_{i,j}$, vertical angle change $△\alpha_{i,j}$, and horizontal angle change $△\beta_{i,j}$ between the target node $N_{j}$ and the reference node $N_{i}$ are normalized, as shown in Formulas (2)–(4).
(2)$$\bar{△{\mathrm{Dist}}_{i,j}}=\frac{\left|{\vec{OA}}_{now}-{\vec{OA}}_{before}\right|}{2{\mathrm{Dist}}_{com}},$$
(3)$$\bar{△\alpha_{i,j}}=\left\{\begin{array}{c} \frac{\left|\alpha_{now}-\alpha_{before}\right|}{180},\;\;\;\;\;\;\;\;\;\;\;\;\;\;\;\;\;\;\;\;\;\left|\alpha_{now}-\alpha_{before}\right|\leq 180{}^{\circ}\;\;\\ \frac{360-\left|\alpha_{now}-\alpha_{before}\right|}{180},\;\;\;\;\;\;\;\;\;\;\;\;\;\left|\alpha_{now}-\alpha_{before}\right|>180{}^{\circ} \end{array}\right.,$$
(4)$$\bar{△\beta_{i,j}}=\left\{\begin{array}{c} \frac{\left|\beta_{now}-\beta_{before}\right|}{180},\;\;\;\;\;\;\;\;\;\;\;\;\;\;\;\;\;\;\;\;\;\;\;\left|\beta_{now}-\beta_{before}\right|\leq 180{}^{\circ}\;\;\\ \;\frac{360-\left|\beta_{now}-\beta_{before}\right|}{180},\;\;\;\;\;\;\;\;\;\;\;\;\;\left|\beta_{now}-\beta_{before}\right|>180{}^{\circ} \end{array}\right.,$$
where ${\mathrm{Dist}}_{com}$ represents the communication distance of the underwater communication node.

Based on the reference node, when the change in the neighboring nodes of the target node is similar to that of the reference node, and the change in the distance between the target node and the reference node, as well as the changes in the vertical and horizontal angles between them, is close to 0, from the perspective of location features, the target node is similar to the reference node in terms of features. Conversely, it is considered that the target node is suspicious.

(2)Node Communication Ability Feature Attribute

The communication ability of a node is reflected in its signaling activity, signal strength, and remaining energy. The signaling activity, also known as the signaling frequency, refers to the number of times a node sends signals per unit time. The signaling activity $Act_{j}$ of the target node $N_{j}$ is obtained by using the ratio of the number of signals $Send_{j}$ sent by the target node per unit time to the number of signals sent $Send_{total}$ by the neighbor nodes monitored by the current node $N_{i}$, as shown in Formula (5).
(5)$$Act_{j}=\frac{Send_{j}}{Send_{total}},$$

If the activity level of the target node is too high or too low, it is considered that there may be an attack on the target node. Therefore, the activity level of the node is selected as a feature attribute of the node’s communication ability.

In the underwater environment covered by the transmission signal, the signal strength of homogeneous nodes mainly decreases with an increase in the transmission distance. The received signal strength exceeding the rated signal strength and abnormal changes can be used as a basis for judging the invasion of abnormal nodes. Therefore, the received signal strength is selected as the feature attribute for judging the trust of the transmitting node. In unit time, the signal strength feature attribute $SP_{j}$ is calculated from the received target node $N_{j}$’s signal transmission strength $Signal_{j}$ and the rated signal transmission strength $Signal_{rated}$, as shown in Formula (6).
(6)$$SP_{j}=\left\{\begin{array}{c} \frac{\left|Signal_{j}-Signal_{rated}\right|}{Signal_{rated}},Signal_{j}\leq 2Signal_{rated}\;\\ \;\;\;\;\;\;\;\;\;\;\;\;\;\;\;1\;\;\;\;\;\;\;\;\;\;,\;\;\;\;\;\;\;\;\;\;Signal_{j}>2Signal_{rated}\; \end{array}\right.,$$

Due to the difficulty in replenishing the energy of underwater nodes, the remaining energy is an important indicator of the ability of nodes to provide continuous communication services. However, excessive remaining energy and abnormal trends in this remaining energy are considered to be suspicious. The remaining energy characteristic attribute $RE_{j}$ is calculated by the current remaining energy $Energy_{j}$ and the rated full energy $Energy_{rated}$ of the target node $N_{j}$ in unit time, as shown in Formula (7).
(7)$$RE_{j}=\left\{\begin{array}{c} 0,\;\;\;\;\;\;\;\;\;\;\;\;\;\;\;\;\;\;\;\;\;\;\;\;\;\;\;\;\;\;\;\;\;Energy_{j}\leq Energy_{rated}\\ \frac{Energy_{j}-Energy_{rated}}{Energy_{rated}},\;Energy_{rated}<Energy_{j}\leq 2Energy_{rated}\\ 1,\;\;\;\;\;\;\;\;\;\;\;\;\;\;\;\;\;\;\;\;\;\;\;\;\;\;\;\;\;\;\;Energy_{j}>2Energy_{rated} \end{array}\right.,$$

When the signal strength and remaining energy of the target node deviate significantly from the rated values, it is considered that the target node is suspicious.

#### 4.1.2. Transmission Channel Trust Feature

In the underwater wireless sensor network, the packet loss rate reflects the quality of the packet forwarding in the transmission channel. By detecting whether the packet loss rate in the transmission channel between the current node and the target node is abnormal, it can be determined whether the identity of the target node is suspicious. When the current node $N_{i}$ sends packets to the target node $N_{j}$, the packet loss rate ${\mathrm{PLR}}_{j}^{i}$ refers to the ratio of the number of packets not forwarded $\left(SP_{i\to j}-FP_{j}^{i}\right)$ by the target node $N_{j}$ to the number of packets sent $SP_{i\to j}$, as shown in Formula (8).
(8)$${\mathrm{PLR}}_{j}^{i}=1-\frac{FP_{j}^{i}}{SP_{i\to j}},$$
where $FP_{j}^{i}$ is the number of packets forwarded by the target node $N_{j}$ and sent by the current node $N_{i}$.

When both the transmitting and receiving nodes are normal nodes, the packet loss rate is only related to adverse factors such as attenuation, multipath, obstruction, and interference in the transmission channel. Within the range of neighboring nodes, the packet loss rate is also similar. Therefore, an abnormal packet loss rate is also used as a basis for determining whether a target node is suspicious.

#### 4.1.3. Data Trust Feature

In underwater wireless sensor networks, the correct packet forwarding by the target node is the goal of the current node selecting the target node as the next-hop node. The current node can determine whether the identity of the target node is suspicious by monitoring and detecting the consistency of the data forwarded by the target node with the original data of the current node. Data consistency can be reflected by the packet error rate. The packet error rate ${\mathrm{PER}}_{j}^{i}$ refers to the ratio of the number of packets that are incorrectly forwarded to the number of packets forwarded, as shown in Formula (9).
(9)$${\mathrm{PER}}_{j}^{i}=\frac{WFP_{j}^{i}}{FP_{j}^{i}},$$
where $WFP_{j}^{i}$ is the number of error packets forwarded by the current node $N_{i}$ to the target node $N_{j}$.

For a normal target node, the data consistency indicator should be as close to 0 as possible and similar to the data consistency indicators of other nodes. Otherwise, the target node is suspicious.

### 4.2. Trust Evaluation Matrix and Model

The trust evaluation matrix that measures the nodes’ trust is constructed through the node trust feature attribute. The trust evaluation matrix of the current node is sent to the node trust evaluation model in a block structure and time period manner for training. The trust evaluation model learns from the trust evaluation matrix of the trusted nodes in a continuous period, establishes the trust feature profile of the trusted nodes, and compares it with the trust evaluation matrix of the node to be evaluated to determine the nodes’ trust. To overcome the sparsity of underwater state interactions, which leads to insufficient model training, the trust evaluation model is initially constructed based on a simulated network using pre-training methods and is updated using transfer learning methods based on actual networks in the application stage.

#### 4.2.1. Trust Evaluation Matrix

In the underwater wireless sensor network, the current node $N_{i}$ evaluates the trust of the target node $N_{j}$ to determine whether it is a malicious node and whether it can be selected in the transmission route. Therefore, it is necessary to first collect the trust status of the current node $N_{i}$, and then evaluate the trust of the target node $N_{j}$ based on the trust feature profile of the current node $N_{i}$. This trust status is constructed according to the trust feature attributes of the node. To reflect the changes in the trust status over time, the trust feature attributes of the target node for five consecutive periods are selected to be included in the elements of the trust evaluation data; to further reflect the gradient relationship of trust feature attribute changes, the difference value of the trust feature attributes for five consecutive periods is also selected to be included in the elements of the trust evaluation matrix.

**Definition** **1.***The trust evaluation matrix of node* 
$N_{j}$*, at time t, is shown in Formula (10)*.
(10)$$TS_{j}^{t}=\left[\vec{TC_{j}^{t}\left[0\right]},\vec{TC_{j}^{t}\left[1\right]},\ldots \vec{,TC_{j}^{t}\left[8\right]}\right],$$
*where* 
$\vec{TC_{j}^{t}\left[0\right]}$ *corresponds to the neighbor node amount change trust feature attribute vector composed of* 
$\Delta Neigh_{j}$*,* 
$\vec{TC_{j}^{t}\left[1\right]}$ *corresponds to the distance change trust feature attribute vector composed of* 
$\bar{△Dist_{i,j}}$*,* 
$\vec{TC_{j}^{t}\left[2\right]}$ *corresponds to the vertical angle change trust feature attribute vector composed of* 
$\bar{△\alpha_{i,j}},\;\vec{TC_{j}^{t}\left[3\right]}$ *corresponds to the horizontal angle change trust feature attribute vector composed of* 
$\bar{△\beta_{i,j}}$*,* 
$\vec{TC_{j}^{t}\left[4\right]}$ *corresponds to the signaling activity trust feature attribute vector composed of* 
$Act_{j}$*,* 
$\vec{TC_{j}^{t}\left[5\right]}$ *corresponds to the signal strength trust feature attribute vector composed of* 
$SP_{j},$ $\vec{TC_{j}^{t}\left[6\right]}$ *corresponds to the remaining energy trust feature attribute vector composed of* 
$RE_{j},$ $\vec{TC_{j}^{t}\left[7\right]}$ *corresponds to the packet loss rate trust feature attribute vector composed of* 
${\mathrm{PLR}}_{j}^{i},$ *and* 
$\vec{TC_{j}^{t}\left[8\right]}$ *corresponds to the packet error rate trust feature attribute vector composed of* 
${\mathrm{PER}}_{j}^{i}$.

The elements of the trust evaluation matrix are shown in Figure 4.

#### 4.2.2. Trust Evaluation Model

The GAN was first proposed in 2014 and consists of two basic networks: a generator and a discriminator. The core logic of GANs is the mutual confrontation and game between the generator and the discriminator. The goal of the generator is to generate data as realistic as possible, preferably to confuse the discriminator and make it unable to distinguish between true and false. Conversely, the goal of the discriminator is to distinguish between true and false as much as possible. GANs are often used for anomaly detection [14]. During training, only normal samples are used to learn the unsupervised manifold distribution in the latent space. During inference, a loss function is defined, and multiple backpropagation iterations are used to find the vector that is closest to the sample distribution in the manifold space. The generator uses forward propagation output to reconstruct normal samples and compares them with the original samples to identify abnormal areas. The output of the discriminator is used for outlier judgment [15]. Therefore, GANs can be used to construct a trust evaluation model for underwater wireless sensor network nodes.

The trust evaluation model constructed based on the GAN is shown in Figure 5. The generator contains two fully connected layers and one ReLU layer, which can generate the trust feature profile of the trusted node based on random data through training. The trust feature profile is composed of 16 trust evaluation matrix blocks. The discriminator contains two fully connected layers, one ReLU layer, and one Sigmoid layer. The discriminator outputs the discrimination probability based on the input trust feature profile of the trusted node. When the discriminative probability is closer to 0.5, this indicates that the trust feature profile of the evaluated node is closer to the trusted node, and the trust of the evaluated node is higher. When the discriminative probability is far away from 0.5 and exceeds the preset malicious node threshold φ, the evaluated node is determined as a malicious node. The malicious node threshold φ is pre-set, which is used to identify the malicious node. It can also be adjusted based on the security level of the forwarded data or adjusted based on the severity of the network topology changes.

The life cycle of the trust evaluation model includes three stages: the initialization training stage, update retraining stage, and evaluation execution stage.

(1)Initialization Training Stage

To overcome the large demand for underwater training data and long duration and minimize the computational energy consumption of underwater nodes for deep learning, a pre-training method in a simulated network is adopted to initialize the trust evaluation model. The trust evaluation model needs to complete the training of both the generator and the discriminator simultaneously. The training process is shown in Figure 6. The generator generates the trust feature profile of the virtual node based on random data. The trust feature profile of the virtual node and the trust feature profile of the trusted node are compared by the discriminator, and the resulting discrimination error further guides the generator to converge, so that the trust feature profile of the “virtual node” generated by it can match the trust feature profile of the trusted node. The discriminator also continuously improves its ability to distinguish between trusted nodes and malicious nodes based on the guidance of the discriminant loss during multiple rounds of training.

The objective function for optimizing the training of the GAN is shown in Formula (11).
(11)$$min_{G}max_{D}V\left(D,G\right)=E_{x\sim P_{trusted}\left(x\right)}\left[\log D\left(x\right)\right]+E_{z\sim P_{z}\left(z\right)}\left[\log\left(1-D\left(G\left(z\right)\right)\right)\right],$$
where x represents the trust state of the trusted node, $P_{trusted}\left(x\right)$ represents the distribution of the trust state of the trusted node, $D\left(x\right)$ is the output score of the discriminator for classifying the trust state of the trusted node, z represents the random data, $P_{z}\left(z\right)$ represents the distribution of the random data, and $G\left(z\right)$ represents the trust state generated by the generator based on the random data. The generator optimizes based on the discriminative loss calculated by $E_{z\sim P_{z}\left(z\right)}\left[\log\left(1-D\left(G\left(z\right)\right)\right)\right]$ to make the generated virtual node’s trust feature profile similar to that of the trusted node so that the discriminator cannot distinguish between them. Formula (11) represents the optimization goal of the discriminator, which is to improve the ability to distinguish between trusted and malicious nodes through a game with the generator.

(2)Update Retraining Stage

When the trust evaluation model is first deployed underwater or after the parameters of the underwater transmission environment change, the discrimination probability generated by the discriminator inputting the trust feature profile generated by the trusted node may deviate from the benchmark value and exceed the preset model deterioration threshold θ, which represents that the trust evaluation model is no longer able to correctly distinguish between trusted nodes and malicious nodes. Then, it is necessary to initiate the parameter update and retraining process of the trust evaluation model. The retraining process of the trust evaluation model is shown in Figure 7. During the retraining process, it is necessary to update the parameters of the existing generator and discriminator based on the data collected in the actual underwater network. Unlike the initialization training stage, the retraining process uses a fine-tuning approach, which only adjusts the parameters of the last fully connected layer of the generator and discriminator. Since the malicious node discrimination ability of the trust evaluation model only deteriorates with the gradual change in the transmission environment parameters, rather than a functional change, the focus of the parameter update should be to adjust the discrimination ability of the trust evaluation model for detailed features of the trust state. Using a fine-tuning approach for the model parameter update not only fully utilizes the training results from the initialization stage and compresses the retraining time, but also reduces the computational energy consumption caused by the model update.

(3)Evaluation and Execution Stage

After the training of the trust evaluation model is completed, the current node periodically conducts trust evaluations on its neighboring nodes according to the trust evaluation model, as shown in Figure 8. If there are more than two trusted neighboring nodes that have evaluated a node to be a malicious node within the period τ, then the node will be directly judged as a malicious node. Then, the trained discriminator will evaluate the target node based on the trust feature profile of the trusted nodes. When the result of the discrimination error is greater than the preset malicious node threshold φ, the node to be evaluated will be determined as a malicious node. Nodes that have passed the trust evaluation are determined as trusted nodes and can be included in the trusted data forwarding nodes set. Nodes that have been determined as malicious nodes are classified as isolated nodes, which only receive status data and do not forward service data. A list of the malicious nodes that have been evaluated during this period will be announced among neighboring nodes.

### 4.3. Process of Trust Evaluation Algorithm

The input of the trust evaluation algorithm is the trust feature attributes of neighboring nodes collected periodically. The trust feature attributes of each neighboring node from multiple periods are assembled into the trust evaluation matrixes, which are then fed into a pre-trained trust evaluation model in batches. The model outputs a judgment on the trust of each neighboring node. Finally, a set of trusted neighboring nodes is formed. The process of the trust evaluation algorithm is defined as Algorithm 1.
**Algorithm 1:** Trust Evaluation AlgorithmNode $N_{i}$ is the current node and is the trusted node.Node $N_{j}$ is one of the neighbor nodes of node $N_{i}$, and is the node to be evaluated.When in the initialization training stage   **For** k steps **do**     Obtain the trust feature attributes of Node $N_{i}$ by (1)–(9) from the simulated network     **Form** the trust evaluation matrix of Node $N_{i}$ by (10)     Sample minibatch of noise samples $\left\{z^{\left(1\right)},\cdots z^{\left(16\right)}\right\}$ from noise prior $P_{z}\left(z\right)$
     Sample minibatch of examples $\left\{x^{\left(1\right)},\cdots x^{\left(16\right)}\right\}$ from the trust feature attributes of the trusted node $P_{trusted}\left(x\right)$, where $x^{\left(m\right)}=TS_{i}^{t+m}$
     Adjust all the parameters of the discriminator D and the generator G in sequence by (11)   The trained discriminator $D_{best}$ and the trained generator $G_{best}$ are obtained**When** in the update retraining stage   **For** k steps do     Obtain the trust feature attributes of Node $N_{i}$ by (1)–(9) from the real network     Form the trust evaluation matrix of Node $N_{i}$ by (10)     Sample minibatch of noise samples $\left\{z^{\left(1\right)},\cdots z^{\left(16\right)}\right\}$ from noise prior $P_{z}\left(z\right)$
     Sample minibatch of examples $\left\{x^{\left(1\right)},\cdots x^{\left(16\right)}\right\}$ from the trust feature attributes of the trusted node $P_{trusted}\left(x\right)$, where $x^{\left(m\right)}=TS_{i}^{t+m}$
     Adjust the last fully connected layer parameters of the discriminator $D_{best}^{\prime }$ and the generator $G{\;}_{best}^{\prime }$ to be updated in sequence by (11)   The updated discriminator $D_{best}$ and the updated generator $G_{best}$ are obtained**When** in the evaluation and execution stage   **While** (true)     Obtain the evaluation result of the trusted neighbor nodes for the node $N_{j}$
     **If** (the times of evaluated as malicious node≥2)      The node $N_{j}$ is the malicious node, at the period τ
     
**Else**
       Obtain the trust feature attributes of node $N_{i}$ and $N_{j}$ by (1)–(9) from the underwater network       Form the trust evaluation matrix $TS_{i}^{t}$ and $TS_{j}^{t}$ of node $N_{i}$ and $N_{j}$ at time t by (10), during the period τ
       Obtain the discriminative probability $DP_{i}$ from the discriminator $D_{best}$ where $x^{\left(m\right)}=TS_{i}^{t+m}$
       **If** ($\left|DP_{i}-0.5\right|\geq \theta$)         The trust evaluation model jumps to the update retraining stage       Obtain the discriminative probability $DP_{j}$ from the discriminator $D_{best}$ where $x^{\left(m\right)}=TS_{j}^{t+m}$
       **If** ($\left|DP_{j}-0.5\right|\geq \varphi$)         The node $N_{j}$ is the malicious node       
**Else**
         The node $N_{j}$ is the trusted node         Include node $N_{j}$ in the neighboring trusted node set     Obtain the trusted nodes set consisting of all trusted neighbor nodes     **Wait** for the next period τ


## 5. Adaptive Routing Algorithm

GTR determines the legitimacy of data sources by the GAN-based trust evaluation model and identifies a trusted nodes set for packet forwarding to improve the security of the routing selection. GTR selects the optimal next-hop node based on historical packet forwarding and the node’s current state, using the reinforcement learning algorithm to improve the adaptability of the routing selection. 

### 5.1. Model of Adaptive Routing Algorithm 

The goal of routing algorithms is to achieve reliable and efficient packet forwarding to trusted nodes. To achieve this goal, it is necessary to ensure that both the source node and the forwarding node are trusted. Therefore, a trusted nodes set for packet forwarding needs to be constructed. The routing process of underwater wireless sensor networks can be defined as a Markov Decision Process (MDP). The selection of each hop node is not only based on the current network state, but also related to the previous packet forwarding state. Therefore, Q-Learning can be used to solve the transmission route selection.

#### 5.1.1. Trusted Data Forwarding Nodes Set

Underwater wireless sensor networks consist of m nodes, which can be defined as:(12)$$N=\left\{n_{1},n_{2},n_{3},\cdots ,n_{m}\right\},$$
where n represents the underwater network nodes and m represents the number of nodes.

The neighbor’s set of node $n_{i}$ is represented by $neigh\left(n_{i}\right)$. Based on the trust evaluation model, the trusted nodes set formed by evaluating the neighbors of node $n_{i}$ can be represented as $trust\left(n_{i}\right)$, also known as the trusted nodes set for packet forwarding. Forwarding service packets between nodes within the trusted nodes set is considered as safe. 

We assume that the packet forwarding direction is from the underwater sensor node to the sink node on the surface. Using $dep\left(n_{i}\right)$ to represent the depth of node $n_{i}$, then the trusted out-degree nodes set $TN_{output}\left(i\right)$ for node $n_{i}$, which is the trusted candidate next-hop nodes set, can be defined as:(13)$$TN_{output}\left(i\right)=\left[n_{j}|dep\left(n_{j}\right)-dep\left(n_{i}\right)\leq 0\right]\cap trust\left(n_{i}\right),$$

The trusted in-degree nodes set $TN_{input}\left(i\right)$ of node $n_{i}$, that is, the trusted data source nodes set, can be defined as:(14)$$TN_{input}\left(i\right)=\left[{n_{j}}^{\prime }|dep\left({n_{j}}^{\prime }\right)-dep\left(n_{i}\right)>0\right]\cap trust\left(n_{i}\right),$$

Node $n_{i}$ obtains a packet from a trusted data source node and then forwards it to a trusted next-hop node according to the routing algorithm.

#### 5.1.2. Reinforcement Learning Routing Algorithm

Reinforcement learning is a computational method for agents to optimize target strategies through interaction with the environment, and is often used to solve Markov decision process (MDP) problems. The agent interacts with the environment in multiple rounds to perceive the state of the environment, decide on the next action, receive the reward, correct the value state, and continuously optimize the quality of the action strategy. The process of reinforcement learning is generally represented by the five-tuple $\left(S,A,R,P,\gamma \right)$. S represents the environment, A represents the action, R represents the reward, P represents the transition probability, and γ represents the discount rate. The agent determines the transition probability P based on the environmental state S and cumulative reward R, generates the action A, and produces a new state $S^{\prime }$.

The communication nodes of underwater wireless sensor networks are defined as intelligent agents, and neighboring intelligent agents form transmission routes through cooperation to achieve packet forwarding.

**Definition** **2.***At time t, if the packet is located at node* 
$n_{i}$*, then the environmental state S can be defined as:*
(15)$$S=\left\{n_{i}\right\}\cup TN_{output}\left(i\right),$$
*At time t, action A can be defined as:*

(16)
$$A=\left\{n_{j}|n_{j}\in S\right\},$$

*At time t, the reward obtained by forwarding the packet from node* $n_{i}$ *to node* $n_{j}$ *can be calculated based on the reward function* *, which is defined as:*(17)$$R_{n_{i}n_{j}}^{a_{j}}=\omega_{d}\times re\left(d\right)+\omega_{e}\times re\left(e\right)+\omega_{o}\times re\left(o\right),$$*The reward* $R_{n_{i}n_{j}}^{a_{j}}$ *includes three parts: depth reward* $\omega_{d}\times re\left(d\right)$*, energy reward* $\omega_{e}\times re\left(e\right)$*, and out-degree node reward* $\omega_{o}\times re\left(o\right)$. $\omega_{d}$*,* $\omega_{e}$*, and* $\omega_{o}$ *are the adjustment coefficients for the depth reward, energy reward, and out-degree node reward, respectively.*(*1*)$re\left(d\right)$*can be defined as:*(18)$$\mathrm{re}\left(d\right)=\frac{dep\left(n_{i}\right)-dep\left(n_{j}\right)}{\underset{n_{k}\in TN_{output}\left(i\right)}{\max}\left(dep\left(n_{i}\right)-dep\left(n_{k}\right)\right)},$$(*2*)$re\left(e\right)$*can be defined as:*(19)$$\mathrm{re}\left(e\right)=\frac{energy_{n_{j}}}{\underset{n_{k}\in TN_{output}\left(i\right)}{\max}energy_{n_{k}}},$$(*3*)$re\left(o\right)$*can be defined as:*(20)$$re\left(o\right)=\frac{num_{output}^{n_{j}}}{\sum_{n_{k}\in N_{relay}\left(i\right)}num_{output}^{n_{k}}},$$*where* $dep\left(n_{j}\right)\;$*represents the depth of node* $n_{j}$*,* $energy_{n_{j}}$ *represents the remaining energy of node* $n_{j}$*, and* $num_{output}^{n_{j}}$ *represents the number of trusted out-degree nodes of node* $n_{j}$.

In the reward function, the purpose of setting the depth reward is to drive the current node to select a shallower next-hop node. The purpose of setting the energy reward is to drive the current node to select a next-hop node with greater remaining energy. The purpose of setting the robustness reward is to drive the current node to select a next-hop node with more outgoing neighbors, so that the packets have more opportunities to be forwarded again, improving the network robustness.

Q-Learning is a type of model-free reinforcement learning algorithm based on value functions, which implements policy optimization based on temporal difference algorithms, and is a typical method for solving Markov Decision Process (MDP) problems.

According to the Q-Learning, at time t, the Q value iteration formula is:(21)$$Q_{i}^{t+1}\left(s_{i}^{t},a_{i}^{t}\right)=\left(1-\alpha \right)Q_{i}^{t}\left(s_{i}^{t},a_{i}^{t}\right)+\alpha \left\{R_{n_{i}n_{j}}^{a_{j}}+\gamma \cdot V_{i}^{t}\left(s_{j}^{t+1}\right)\right\}\;,$$
where α is the learning rate, which reflects the learning of historical forwarding experience by policy optimization.

According to the Bellman equation, the value function at time t is:(22)$$V^{t}\left(s\right)=\underset{a}{\max}Q^{*}\left(s,a\right)\;\;,$$

In the early stage of reinforcement learning, the Q value corresponding to each transmission policy is assigned an initial value. As learning progresses, the Q value is continuously updated according to Formulas (21) and (22) and converges, and a good transmission routing strategy is eventually highlighted.

### 5.2. Packet for Adaptive Routing Algorithm

The state and control information involved in GTR is implemented through packets forwarded between nodes. The packet of GTR includes two types: service packets and control packets. Service packets are forwarded in the service channel, mainly carrying monitoring data. The forwarding path is transmitted hop-by-hop from the underwater sensor node to the surface sink node through the communication node, and the forwarding range is the set of trusted nodes. The goal of GTR is to provide secure and adaptive transmission routes for service packets. Control packets are forwarded in the control channel, mainly used to synchronize the status of neighboring nodes. After receiving them, the neighboring nodes do not forward the control packets, and the forwarding path is limited to the current node and its neighboring nodes. Each node updates the status of the current node and neighboring nodes in the database by listening to the service packets and control packets.

Service packets are generated by the current sending node and sent in a broadcast form, which can be received by all neighboring nodes. After receiving the packet, the receiving node correctly decodes the received data and forwards the packet according to the forwarding requirements. Other neighbor nodes discard the packet after obtaining the status information in the packet. The composition structure of a service packet is shown in Figure 9a. The first and second fields are the node IDs of the sending node and receiving node. The third field is the status function value forwarded by the current node to the receiving node, that is, the V value. The fourth and fifth fields are the timestamp of the packet being forwarded and the number of sending times it has been repeatedly forwarded. The sixth and seventh fields are the sequence number and content of the forwarded data. When the receiving node feeds back the receiving result, it sets the V value in the third field to 255, indicating successful data reception, and updates the sending timestamp to the sending timestamp of the receiving node.

Control packets are broadcasted by each node to their neighboring nodes at regular intervals. The neighboring nodes update their local information databases based on the control packet fields. The composition structure of a control packet is shown in Figure 9b. The first to third fields are the current node ID, location, and remaining energy. The fourth and fifth fields are the current node’s trusted data forwarding nodes set, which includes the number of trusted in-degree nodes and out-degree nodes. The sixth field is the malicious neighbor nodes set, which includes the number of malicious neighboring nodes and all malicious node IDs.

### 5.3. Process of Adaptive Routing Algorithm

The current node receives the packet from the trusted data source nodes and forwards it by adaptive routing algorithms. The input of the adaptive routing algorithm is the state of the candidate next-hop nodes. It calculates the Q value of each candidate next-hop node based on Q-Learning and selects the trusted neighboring node with the highest Q value as the next-hop node. Then, the current node forwards the packet to the next-hop node. The process of the adaptive routing algorithm is defined as Algorithm 2.
**Algorithm 2:** Adaptive Routing Algorithm**While** (true)   **If** (the new packet arrived)      Update the status of neighboring nodes in the database of the current node   **If** (the new service packet from the trusted data source nodes to forward)      **While** ($times_{send}\leq tims_{max}$)        Update the S and A by (15)–(16)        Calculate the $R_{n_{i}n_{j}}^{a_{j}}$ of each action by (17)–(20)        Calculate the $Q_{i}^{t+1}\left(s_{i}^{t},a_{i}^{t}\right)$ of each action by (21)        Update the V-value with the max Q value using Formula (22)        Forward the packet to the next-hop node with the max Q value        **If** (It is detected that the packet has been forwarded)         
**Break**
        
**Else**
         ${\mathrm{times}}_{\mathrm{send}}$++
where $times_{send}$ is the times of retransmissions and $tims_{max}$ is the threshold for the maximum times for retransmissions.

## 6. Performance Evaluation

The performance evaluation experiment mainly focuses on discussing the performance of GTR in terms of trust evaluation and packet forwarding under different malicious node ratios, attack mode switching intervals, newly added unknown malicious nodes, and different node location dynamic ranges and communication interruption probability conditions.

### 6.1. Experimental Environment

The experiment is based on underwater monitoring applications. The experimental data come from the underwater temperature observation data of the KEO station in the NOAA database. The environment for the experiment is set to a three-dimensional underwater area of 500 m × 500 m × 500 m. A sensor node is deployed at the center of the underwater area to collect data, and 100 communication nodes are randomly deployed in the water to monitor and forward packets in real time. A sink node is deployed at the center of the surface area to collect data and transmit them to the data center. Each set of monitoring data occupies 2 bytes, and each packet encapsulates five sets of monitoring data for forwarding. The underwater sensor node, communication nodes, and sink node all use underwater acoustic communication. The communication and sensing distance of the underwater nodes is set to 150 m. Underwater nodes move randomly around their original location due to the influence of water flow, with maximum movement distances set to 0, 5, 10, 15, and 20 m. Underwater sensor nodes and sink nodes have external energy supplies and do not consider energy consumption. The communication node is powered by a battery, with an initial energy of 1000 J, a transmission energy consumption of 10 W, a reception energy consumption of 2 W, and a calculated energy consumption of 0.2 W. The communication interruption probability caused by noise interference, multipath interference, and occlusion interference between nodes can be set to 0, 5%, 10%, 15%, and 20%. The sound frequency is selected to be 10 kHz, and the underwater propagation speed of sound is set to 1500 m/s. The S-FAMA is used to implement the MAC layer protocol and divide the transmission channel. The proportion of malicious nodes in the network can be selected to be 5%, 10%, 15%, 20%, and 30%, with random switching between normal and eight attack modes. The attack mode switching interval is set to no change, 10 min, 5 min, 2 min, and 1 min. Underwater networks may be mixed with malicious nodes with unknown attack patterns, with proportions of 0, 5%, 10%, 20%, and 30% of malicious nodes. The environment is built based on Python, and GTR is implemented using PyTorch. The relevant hyperparameters of GTR are shown in Table 3.

### 6.2. Performance Analysis of Trust Evaluation

The goal of the underwater wireless sensor network node trust evaluation is to correctly distinguish between trusted nodes and malicious nodes and ensure the security of their transmission routes. This experiment evaluates the performance of the trust evaluation model with the detection rate, error detection rate, and AUC (area under the curve) value of the ROC curve. The detection rate, also known as the true positive rate (TPR), refers to the ratio of the number of times a malicious node is correctly detected to the total number of detections. The error detection rate, also known as the false positive rate (FPR), refers to the ratio of the number of times a trusted node is erroneously detected as a malicious node to the total number of detections. Evaluation models with a high TPR and low FPR have a better performance, while those with a lower TPR and higher FPR have a worse performance. The ROC curve reflects the performance changes in the trust evaluation model at different thresholds and is plotted with the TPR as the vertical coordinate and the FPR as the horizontal coordinate. It can more comprehensively evaluate the generalization ability and stability of the trust evaluation model. The AUC is the area under the ROC curve. The larger the AUC value, the better the performance of the trust evaluation model, and vice versa. The trust evaluation performance of GTR will be compared with that of the TEUC and DOIDS in the following aspects. In the trust evaluation experiment, GTR, TEUC, and DOIDS are all pre-trained models. In the underwater network, both trusted nodes and malicious nodes are randomly assigned based on the experimental conditions. The TPR, FPR, and ROC-AUC values are calculated based on the statistics for the situation of all the trusted nodes detecting whether their neighboring nodes are malicious nodes during 50 periods.

#### 6.2.1. Different Proportions of Malicious Nodes

The experiments based on different malicious node proportions refer to maintaining the total number of nodes in the underwater network as unchanged, adjusting the proportions of malicious nodes in the nodes to 5%, 10%, 15%, 20%, and 30% and conducting trust evaluation performance experiments for GTR, TEUC, and DOIDS. Figure 10 shows the changes in the TPR, FPR, and ROC-AUC values of GTR, TEUC, and DOIDS under different malicious node ratios.

As shown in Figure 10, GTR maintains a higher TPR and lower FPR compared to TEUC and DOIDS, and its ROC-AUC value also remains at a high level. When the proportion of malicious nodes is 5% or 10%, TEUC has the best trust evaluation performance due to its use of a supervised C45 algorithm and learning from labeled training samples. Although GTR uses an unsupervised learning method, its trust evaluation performance can approach that of TEUC when the proportion of malicious nodes is 5% or 10%. As the proportion of malicious nodes increases, the TPRs of GTR, TEUC, and DOIDS all decrease, the FPRs all increase, and the ROC-AUC values all decrease. This is due to the feature of the newly added malicious nodes exceeding the resolution capability of the trust evaluation model. However, GTR’s degradation rate is more gradual than that of TEUC and DOIDS, and its evaluation performance jumps to the top after the proportion of malicious nodes exceeds 15%. This is because GTR mainly focuses on the trust features of trusted nodes. Although the increase in the proportion of malicious nodes and the diversification of malicious node trust features blur the classification boundary to some extent, the overall change in the trust features of the trusted nodes is relatively stable. As the proportion of malicious nodes changes, DOIDS’ evaluation performance deteriorates more rapidly, which is due to DOIDS being implemented based on the unsupervised DBSCAN algorithm. A change in the malicious node density leads to a decrease in DOIDS’ clustering quality.

#### 6.2.2. Different Attack Mode Switching Intervals

The experiments are based on adjusting the attack modes of the malicious nodes at intervals of unchanged, 10 min, 5 min, 2 min, and 1 min under the condition that the proportion of malicious nodes is 15%. Figure 11 shows the changes in the TPR, FPR, and ROC-AUC values of GTR, TEUC, and DOIDS under different attack mode switching intervals.

As shown in Figure 11, as the attack mode switching interval decreases, the performances of the GTR, TEUC, and DOIDS in terms of trust evaluation slightly decrease. When the switching interval reaches 5 min, the performance of the trust evaluation decreases significantly. The trust evaluation of GTR comprehensively considers the trust feature of nodes in multiple consecutive periods to improve its evaluation performance. However, the rapid switching of attack modes can cause the overlapping of multiple attack mode features, resulting in the blurring of classification boundaries. Since GTR mainly relies on the trust feature profile of trusted nodes for classification, the ambiguity of the malicious node attack features has a relatively small impact on GTR. Therefore, the performance degradation of GTR is relatively smooth. As the attack mode switching interval decreases, the frequency requirement for evaluating trust increases. When the evaluation frequency cannot match the attack mode switching interval, it will have a significant impact on malicious node detection.

#### 6.2.3. Newly Added Unknown Malicious Nodes

The experiments based on newly added unknown malicious nodes are conducted under the condition that malicious nodes with unknown attack patterns are added, with proportions of 0, 5%, 10%, 20%, and 30%. Figure 12 shows the changes in the TPR, FPR, and ROC-AUCs value of the trust evaluation performances of GTR, TEUC, and DOIDS under different proportions of newly added unknown malicious nodes.

As shown in Figure 12, with the increase in the proportion of unknown malicious nodes, the performances of GTR, TEUC, and DOIDS in evaluating trust decline. Among them, the performance of TEUC declines the most. Because TEUC uses the supervised learning algorithm C4.5 to achieve trust evaluation, it lacks a classification branch for unknown malicious nodes. It cannot correctly identify unknown malicious nodes, resulting in a significant performance decline. Both GTR and DOIDS use unsupervised algorithms to achieve trust evaluation, which can reduce the impact of lacking trust feature data for unknown malicious nodes to some extent. However, DOIDS implements trust evaluation based on the DBSCAN density clustering algorithm, which can cause ambiguity in the boundary of the trust evaluation when the neighborhood of unknown malicious nodes cannot reach the density of known malicious nodes, affecting the performance of DOIDS. Therefore, as the proportion of unknown malicious nodes increases, the performance of DOIDS in evaluating trust also shows a certain degree of decline. The trust evaluation ability of GTR mainly comes from learning the trust feature profile of trusted nodes. Although an increase in the proportion of unknown malicious nodes blurs the trust feature profile of trusted nodes to some extent, the overall impact of this is insignificant. Therefore, as the proportion of unknown malicious nodes increases, the performance of GTR in evaluating trust does not significantly decline compared to TEUC and DOIDS.

#### 6.2.4. Different Dynamic Node Location Range

The experiments based on nodes with different dynamic location ranges refer to simulating the different effects of water flow on nodes, setting the dynamic ranges of node location movements to 0 m, 5 m, 10 m, 15 m, and 20 m and conducting trust evaluation performance experiments for GTR, TEUC, and DOIDS. Figure 13 shows the changes in trust evaluation performance for GTR, TEUC, and DOIDS under different node dynamic location ranges.

As shown in Figure 13, with the increase in the dynamic range of the node location, the performances of GTR, TEUC, and DOIDS in evaluating the trust of nodes decrease, especially with an increase in the FPR. Due to the increase in the dynamic range of the node location, the classification boundary between the trusted nodes and malicious nodes is further blurred, resulting in some trusted nodes being misclassified as malicious nodes. GTR uses multiple periods of trust feature attributes and their differences as the trust feature profile of trusted nodes. The continuity of the location movements of trusted nodes over time reduces the impact of dynamic changes in node locations. GTR uses a method of comparing the trust feature profiles of the current node and its neighboring nodes to distinguish malicious nodes, which makes the correlation of location movements between neighboring nodes in space reduce the impact of the increase in the dynamic range of the node location.

#### 6.2.5. Different Inter-Node Communication Interruption Probabilities

The experiments based on different inter-node communication interruption probabilities refer to the simulation of underwater communication using acoustic signals affected by multipath, obstruction, and noise, resulting in underwater communication with interruption probabilities of 0, 5%, 10%, 15%, and 20%. Figure 14 shows the changes in the TPR, FPR, and ROC-AUC values of GTR, TEUC, and DOIDS under different inter-node communication interruption probabilities.

As shown in Figure 14, as the inter-node communication interruption probability increases, the trust evaluation performances of GTR, TEUC, and DOIDS all decrease, especially with an increase in the FPR. Due to the increase in the inter-node communication interruption probabilities, the classification boundary between trusted nodes and malicious nodes is further blurred, resulting in some trusted nodes being misclassified as malicious nodes. Especially when the communication interruption probability is greater than 10%, it has a significant impact on various models. Changes in the communication interruption probabilities between nodes are usually caused by sudden changes due to noise, occlusion, and marine environmental effects. GTR uses a trigger-based model update method, which triggers GTR to retrain when it detects a change in the communication interruption probability between the current node and its neighbors. The transmission channel between the current node and its neighbors is generally similar, which can minimize the misclassification caused by sudden changes in the communication interruption probabilities between nodes. Therefore, as the communication interruption probabilities between nodes increase, GTR has the slowest degradation in the trust assessment performance compared to TEUC and DOIDS.

### 6.3. Performance Analysis of Data Forwarding 

The goal of the underwater wireless sensor network routing algorithm is to achieve secure, efficient, and reliable data forwarding from underwater sensor nodes to surface sink nodes through underwater communication nodes. The data forwarding performance is mainly reflected in three aspects: the packet delivery rate, energy tax, and network throughput. GTR will be compared with QELAR and DOIDS for its data forwarding performance.

This performance comparison will be conducted based on different malicious node ratios, different attack mode switching intervals, different proportions of newly added unknown malicious nodes, different dynamic ranges of node location, and different inter-node communication interruption probabilities. The scenario settings for this performance analysis of data forwarding are shown in Table 4. In the same scenario, the underwater network topology configured by GTR, QELAR, and DOIDS is consistent. Packets are sent by sensor nodes located underwater, through underwater communication nodes, and finally delivered to sink nodes located on the surface.

#### 6.3.1. Comparison of Packet Delivery Rate

The packet delivery rate refers to the ratio of the number of packets received by the surface sink node to the number of packets sent by the underwater sensor node. A higher packet delivery rate reflects that the routing algorithm can avoid the influence of malicious nodes to the maximum extent and deliver packets to the destination in a safer way. Figure 15 shows a comparison of the packet delivery rates for sending 100 packets in different scenarios.

It is not difficult to see from Figure 15 that the packet delivery rates of DOIDS, QELAR, and GTR are the highest in scenario I, and the gap between them is also the smallest. In scenario II, due to the addition of malicious nodes, the packet delivery rates of DOIDS, QELAR, and GTR decrease, but QELAR has the largest decrease due to its lack of a malicious node detection mechanism. In scenario III, due to the switching of the malicious node attack modes, the packet delivery rates of DOIDS, QELAR, and GTR decreased compare to scenario II, among which, GTR has the slowest degradation rate. In scenario IV, due to the increase in the proportion of newly added unknown malicious nodes, the packet delivery rates of DOIDS, QELAR, and GTR decrease compared to scenario II, among which, GTR has the highest packet delivery rate, followed by DOIDS and QELAR. In scenario V, the maximum dynamic range of node location changes increases, which blurs the classification boundary between trusted nodes and malicious nodes, as well as the relationships between neighboring nodes. Therefore, the packet delivery rates of DOIDS, QELAR, and GTR decrease compared to scenario II. In scenario VI, the increase in the communication interruption probability also leads to the blurring of the classification boundary between trusted nodes and malicious nodes, as well as the deterioration of transmission channels. The packet delivery rates of DOIDS, QELAR, and GTR decrease compared to scenario II, among which, DOIDS has the largest decrease, followed by QELAR and GTR.

Overall, the packet delivery rate of GTR compared to QELAR and DOIDS remains within a high and stable range and is less affected by various adverse conditions, maintaining a good robustness. This is mainly due to the robustness of GTR, which can maintain a high TPR and low FPR for malicious node detection under various conditions. In addition, because GTR is a reinforcement-learning-based routing algorithm, it can adjust the next-hop node promptly based on historical packet forwarding, thus maintaining a high packet delivery rate.

#### 6.3.2. Comparison of Energy Tax

Energy tax refers to the average energy consumption after allocating the energy consumption required to forward any packet from the underwater sensor node to the surface sink node, as shown in Formula (23), including the receiving energy consumption, the sending energy consumption, and the computational energy consumption of the node. The lower the energy tax, the more efficient the routing algorithm. Figure 16 shows a comparison of the energy tax for sending 100 packets in different scenarios.
(23)$$\mathrm{EnergyTax}=\frac{E_{comsumed}}{m\times R_{packets}}\;,$$
where $E_{comsumed}$ is the total amount of energy consumed by the underwater network for sending 100 packets, m is the number of nodes, which is 100, and $R_{packets}$ is the number of packets sent, which is 100.

It is not difficult to see from Figure 16 that, in scenario I, the energy tax of QELAR is the lowest, followed by GTR, while the energy tax of DOIDS is relatively high. In scenario II, with the addition of malicious nodes, the energy tax of QELAR increases, and the energy tax of GTR and DOIDS both increase, but this increase is relatively smooth. In scenario III, with the switching of the malicious node attack modes, the energy taxes of QELAR, DOIDS, and GTR all slightly increase. In scenario IV, with the addition of unknown malicious nodes, the energy taxes of QELAR, DOIDS, and GTR all increase, among which, the increase for QELAR is obvious. In scenario V, with the increase in the maximum dynamic range of node location changes, the energy taxes of QELAR, DOIDS, and GTR all increase, and the energy tax of GTR compared to scenario II increases. In scenario VI, with the increase in the node communication interruption probability, the energy taxes of QELAR, DOIDS, and GTR all increase, as in scenario V.

Overall, GTR has the lowest energy tax compared to QELAR and DOIDS. Although the energy tax of GTR in scenario I is slightly higher than that of QELAR due to the computational energy consumption caused by the trust evaluation, with the addition of malicious nodes, the communication energy consumption caused by multiple retransmissions in QEALR greatly exceeds the computational energy consumption, resulting in a rapid increase in energy tax from scenario II. DOIDS is classified based on density, which also requires relatively large computational resources. As the maximum dynamic range of node location changes and the communication interruption probability increase, the classification boundary between trusted nodes and malicious nodes becomes further blurred, resulting in a decrease in the TPR and a significant increase in the energy tax. GTR achieves the characterization of the trust feature profile of trusted nodes based on the GAN and uses the trust feature profile of these trusted nodes to distinguish malicious nodes, improving the robustness of the trust evaluation algorithm. The triggered model update method ensures the stability of the detection rate of the trust evaluation. In selecting the next-hop node, it uses Q-Learning to optimize the next-hop node according to the minimum hop count, energy balance, and other optimization objectives, reducing the energy tax required for data forwarding.

#### 6.3.3. Comparison of Network Throughput

Network throughput refers to the total number of packets forwarded from underwater sensor nodes to surface sink nodes hop-by-hop within the lifetime of the underwater wireless sensor network. The lifetime of underwater networks begins with the sensor node sending the first packet and ends with the sink node receiving the last packet. The larger the network throughput, the stronger the routing algorithm’s ability to overcome malicious node attacks, and the more persistent its ability to provide data forwarding services. Figure 17 shows a comparison of the network throughput within the lifetime of the underwater wireless sensor network.

It is not difficult to see from Figure 17 that, in scenario I, the network throughputs of QELAR and GTR are larger than that of DOIDS. In scenario II, with the addition of malicious nodes, the network throughputs of QELAR, DOIDS, and GTR all decrease, but the network throughput of QELAR decreases significantly. In scenario III, with the switching of the malicious node attack modes, the network throughputs of QELAR, DOIDS, and GTR slightly decrease, but this overall change is not significant. In scenario IV, with the addition of unknown malicious nodes, compared to scenario II, the network throughput of GTR decreases slightly, followed by a decrease in the network throughput of DOIDS, while the decrease in the network throughput of QELAR is the most significant. In scenarios V and VI, with increases in the maximum dynamic range of node location changes and communication interruption probability, the network throughputs of QELAR, DOIDS, and GTR all decrease. Although the GTR has a more significant decrease than in scenario II, the overall network throughput is still much higher than that of QELAR and DOIDS.

Network throughput is related to the network lifetime and packet delivery rate, and the network lifetime is related to the energy tax. An algorithm with a lower energy tax and a higher packet delivery rate will also have a higher network throughput. GTR implements the discrimination of malicious nodes based on a trust evaluation model and removes these malicious nodes from the next-hop nodes set, narrowing the action space of data forwarding and improving the efficiency of the route search. This avoids the retransmission energy consumption caused by forwarding packets to malicious nodes. Therefore, compared to the QELAR, GTR can provide a greater network throughput. Through the performance analysis of the trust evaluation, it was not difficult to find that GTR could provide a higher TPR than DOIDS, so it can more accurately remove malicious nodes from the transmission route. The routing selection method based on Q-Learning used by GTR can make more balanced use of the remaining energy between neighboring nodes. The routing selection method based on sending nodes used by GTR is more specific than opportunistic routing, avoiding possible conflicts caused by different next-hop nodes competing for resources. Therefore, GTR has a higher network throughput than DOIDS, which is consistent with the results of the simulation experiments.

## 7. Discussion

In an open and harsh underwater environment, the nodes of underwater wireless sensor networks may suffer from various attacks such as black hole attacks, wormhole attacks, sink hole attacks, Sybil attacks, selective forwarding attacks, exhaustion attacks, jamming attacks, and acknowledgment spooling attacks implemented by malicious nodes, which pose a huge threat to the security of packet forwarding. One of the important methods for addressing security threats is to not receive data from malicious nodes, not forward data to malicious nodes, and exclude malicious nodes from the transmission route. GTR uses a GAN to evaluate the trust of neighboring nodes based on the trust feature profile of the current node (also considered as a trusted node), forming a trusted nodes set for packet forwarding. Within the trusted nodes set, a transmission route is constructed based on Q-Learning. GTR effectively enhances the ability of the transmission route to resist network attacks from various malicious nodes by combining the trust evaluation algorithm with the routing algorithm and enhances the security of packet forwarding. Since packets are only forwarded between trusted nodes, it avoids retransmissions and packet losses or errors due to forwarding to malicious nodes, thus, effectively improving the security, reliability, and efficiency of packet forwarding. The routing algorithm is implemented based on reinforcement learning algorithms, and the optimization objectives fully consider historical forwarding experience, as well as the location, remaining energy, and topology structure of the next-hop node. Therefore, it can obtain a more optimal transmission routing policy, select a more suitable next-hop node, balance the allocation of remaining energy, and extend the service time of the network.

However, there are still some aspects of the proposed algorithm, GTR, that need further improvement in future research. There are two preset hyperparameters in GTR, namely, the malicious node threshold φ and the model degradation threshold θ. In specific scenarios, finding the best two threshold values is very difficult. In most cases, these two thresholds still need to be dynamically adjusted according to the deployment networks and service applications. I think that studying how the two preset thresholds can be set will be an important work direction. I plan to combine fuzzy logic to solve the problem. In research, there is an initial assumption that all nodes in the network are homogeneous. Therefore, the situations of neighboring nodes can be judged based on the trust feature profile of the current node. However, when an underwater network contains different types of network nodes, the proposed method becomes ineffective. Therefore, it is necessary to establish a trust feature profile library for different types of network nodes, and then apply this in conjunction with the knowledge graph.

## 8. Conclusions

An underwater wireless sensor network equipped with GTR can provide more secure, reliable, and efficient packet forwarding services. GTR includes a trust evaluation algorithm and an adaptive routing algorithm. GTR defines trust feature attributes based on nodes, transmission channels, and data, and defines trust feature profiles based on multi-period trust feature attributes and their differential values. It uses a GAN-based trust evaluation model to distinguish malicious nodes. It implements timely updates of the trust evaluation model based on the triggering method. GTR constructs a transmission route based on Q-Learning, optimizes the next-hop node based on the node location, network topology, and remaining energy, and forms a trusted forwarding nodes set based on the evaluation results of the trust evaluation model. It eliminates malicious nodes on the transmission route and reduces the action space for routing selection. Simulation experiments showed that GTR has a high TPR and low FPR. Under the condition of 15% malicious nodes and 10% unknown malicious nodes mixed in, the TPR increased by 5.4% and the FPR decreased by 36.4% compared to baseline algorithms (TEUC and DOIDS). GTR can achieve secure, reliable, and efficient packet forwarding in underwater networks with malicious nodes. Under the condition of 15% malicious nodes and 10% unknown malicious nodes mixed in, the packet delivery rate increased by 11.0%, the energy tax decreased by 11.4%, and the network throughput increased by 20.4% compared to baseline algorithms (QELAR and DOIDS). GTR can be configured for underwater networks used to transmit critical information such as target detection, disaster warnings, etc., to limit malicious nodes from disrupting the transmission security.

## Figures and Tables

**Figure 1 sensors-24-04879-f001:**
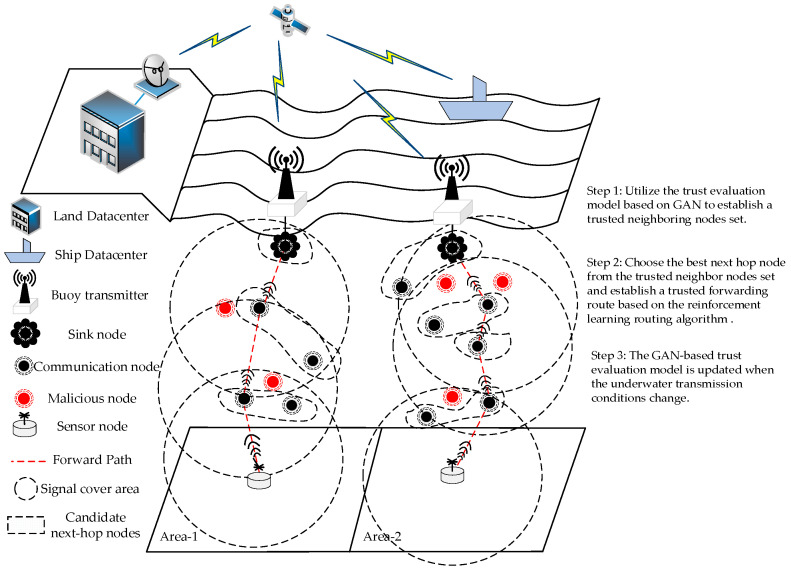
Schematic diagram of the trusted routing model of underwater wireless sensor networks.

**Figure 2 sensors-24-04879-f002:**
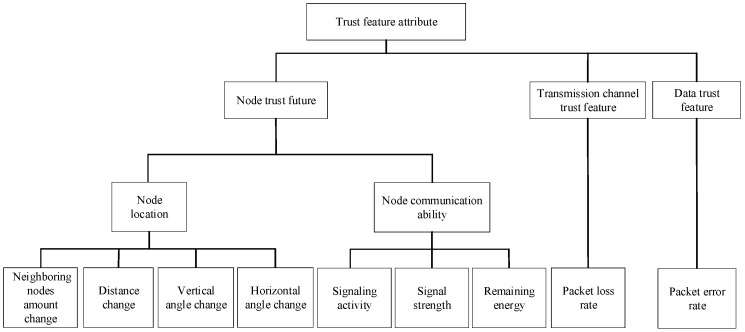
Trust feature attributes.

**Figure 3 sensors-24-04879-f003:**
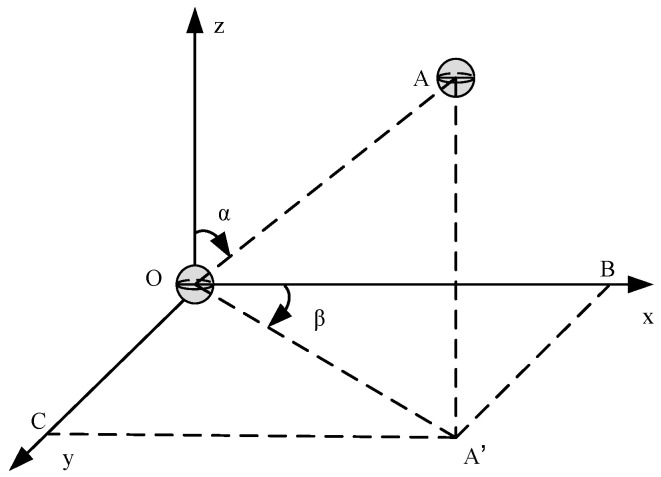
Diagram of node location relationship.

**Figure 4 sensors-24-04879-f004:**
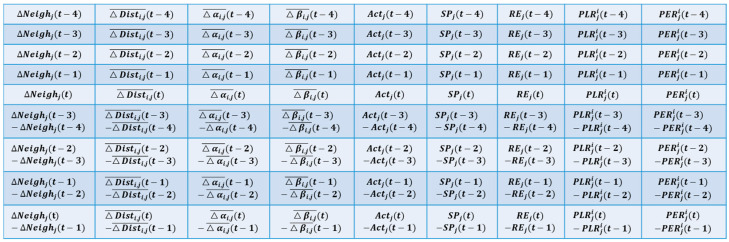
Elements of the trust evaluation matrix.

**Figure 5 sensors-24-04879-f005:**
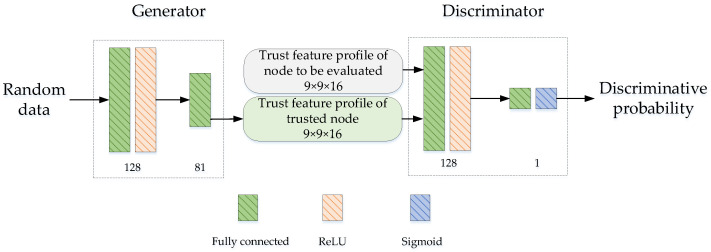
Neural network structure of trust evaluation model.

**Figure 6 sensors-24-04879-f006:**
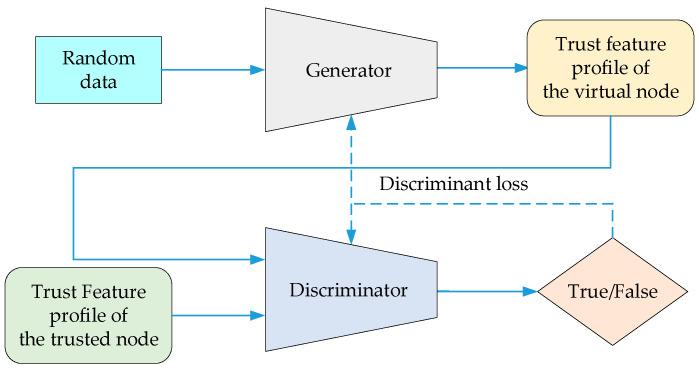
Schematic diagram of the initialization training stage of the trust evaluation model.

**Figure 7 sensors-24-04879-f007:**
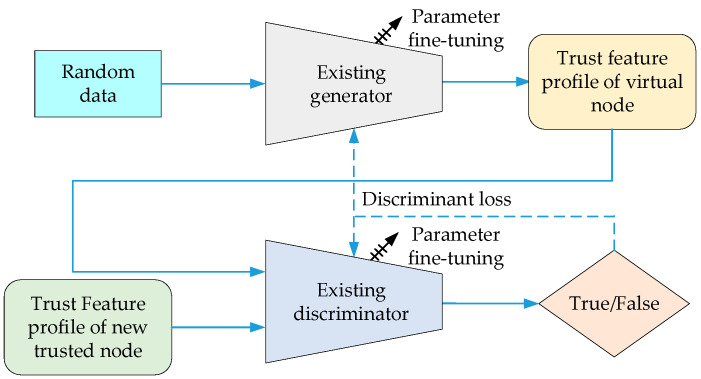
Schematic diagram of the update retraining stage of the trust evaluation model.

**Figure 8 sensors-24-04879-f008:**
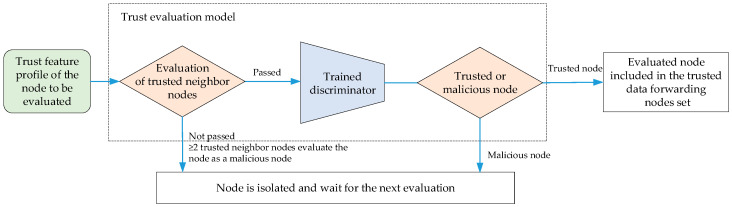
Schematic diagram of the evaluation and execution stage of the trust evaluation model.

**Figure 9 sensors-24-04879-f009:**
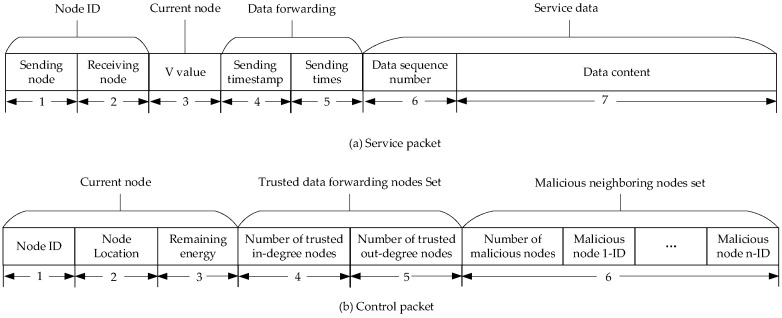
Packet for adaptive routing algorithm.

**Figure 10 sensors-24-04879-f010:**
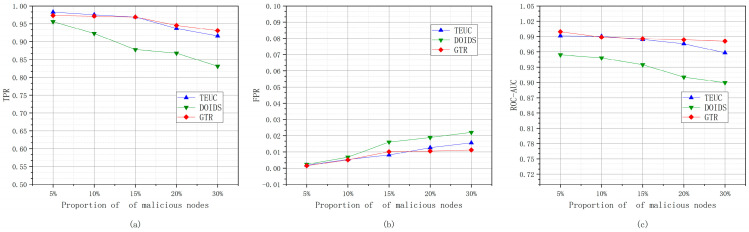
The changes in TPR, FPR, and ROC-AUC under different malicious node ratios.

**Figure 11 sensors-24-04879-f011:**
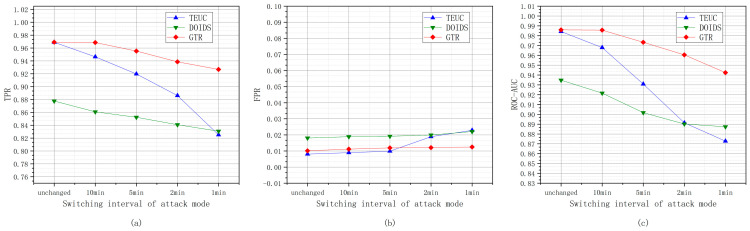
The changes in TPR, FPR, and ROC-AUC under different attack mode switching intervals.

**Figure 12 sensors-24-04879-f012:**
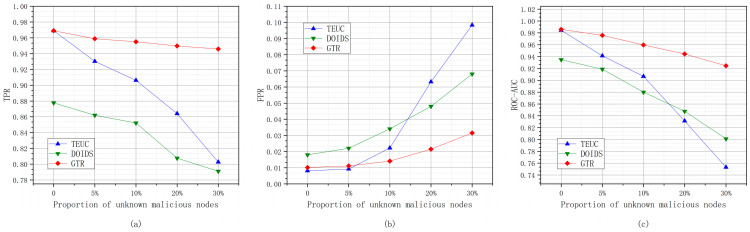
The changes in TPR, FPR, and ROC-AUC under different proportions of newly added unknown malicious nodes.

**Figure 13 sensors-24-04879-f013:**
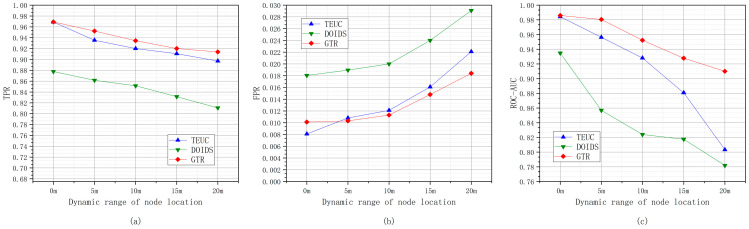
The changes in TPR, FPR, and ROC-AUC under different dynamic node location ranges.

**Figure 14 sensors-24-04879-f014:**
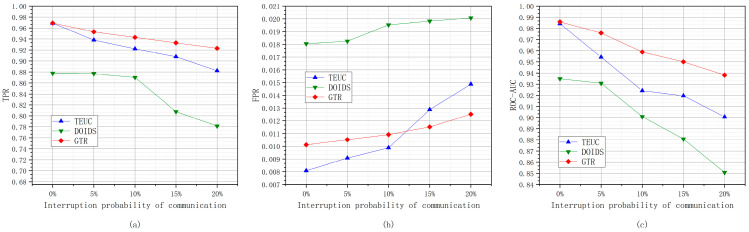
The changes in TPR, FPR, and ROC-AUC under different inter-node communication interruption probabilities.

**Figure 15 sensors-24-04879-f015:**
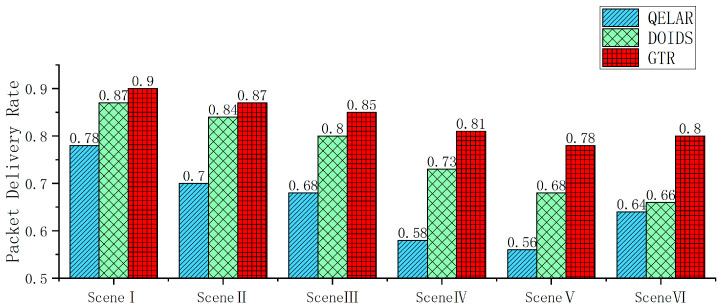
Comparison of the packet delivery rate for sending 100 packets.

**Figure 16 sensors-24-04879-f016:**
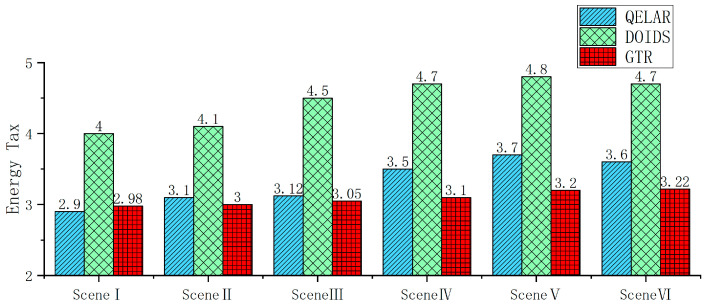
Comparison of the energy tax for sending 100 packets.

**Figure 17 sensors-24-04879-f017:**
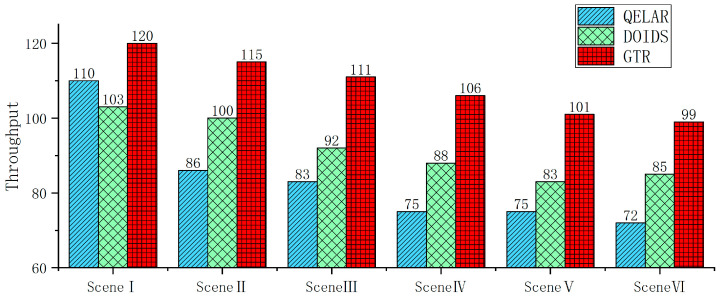
Comparison of the network throughput within the lifetime of the underwater wireless sensor network.

**Table 1 sensors-24-04879-t001:** Several typical secure routing algorithms for underwater wireless sensor networks.

Algorithm	Method of Route Establishment	Design of Secure Packet Forwarding
QELAR, DQIR, QTAR, DROR	Reinforcement learning routing	Determining the route based on packet forwarding experience
SEEORVA, SecFUN, RSN^2^, R-CARP	Opportunistic routing, channel-aware routing, multi-path routing	Employing encryption and signature algorithms to ensure the security of packet forwarding
SARP, ST-CJ, AFSA-ACOA-SC	Beam control routing, intelligent routing	Ensuring packet forwarding security based on topology information
SEECR, DOIDS	Cooperative routing, opportunistic routing	Feedback information, Clustering algorithm
GTR	Reinforcement learning routing	Trust evaluation

**Table 2 sensors-24-04879-t002:** Several typical trust evaluation algorithms for underwater wireless sensor networks.

Algorithm	Evaluation Indicators	Classification Method	Update Method
ARTMM	Node, link, data status	Fuzzy logic, direct calculation	Time window, forgetting factor
TEUC, STMS, TMIS	Node, link, data status	C4.5, SVM	Periodic update
SDFTM	Node, link, data status	SVM, DS evidence	Periodic update
TUMRL	Node, link, data status, criticality of nodes	Reinforcement learning	Periodic update
ITrust	Node, link, data status, environmental trust	Random forest algorithm	Periodic update
LTrust	Node, link, data status, recommendation of neighbor nodes	LSTM	Periodic update
DOIDS	Node, link, data status	DBSCAN	Trigger update
GALTM	Node, link, data status	GAN	Trigger update
GTR	Node, link, data status	GAN	Trigger update

**Table 3 sensors-24-04879-t003:** Hyperparameters of GTR.

Parameter	Value	Explain
φ	0.1	Malicious node threshold
θ	0.05	Model deterioration threshold
$\omega_{d},\;\omega_{e},\omega_{o}$	1, 0.2, and 0.2	Adjustment coefficients for depth reward, energy reward, and out-degree node reward
α, γ	0.8 and 0.5	Learning rate and discount rate
β	1 min	Interval for malicious node detection
$rate_{forwarding}$	2 packets/min	Packet forwarding rate
$tims_{max}$	2	Maximum times for retransmissions

**Table 4 sensors-24-04879-t004:** Scenario settings for performance analysis of data forwarding.

Scenario	Malicious Node Ratio	Attack Mode Switching Interval	Added Unknown Malicious Nodes	Dynamic Range of Node Location	Communication Interruption Probability
Scenario I	$M_{p}=0\%$	$S_{f}=unchanged$	$N_{p}=0\%$	$P_{d}=0$	$O_{p}=0\%$
Scenario II	$M_{p}=15\%$	$S_{f}=unchanged$	$N_{p}=0\%$	$P_{d}=0$	$O_{p}=0\%$
Scenario III	$M_{p}=15\%$	$S_{f}=2$ min	$N_{p}=0\%$	$P_{d}=0$	$O_{p}=0\%$
Scenario IV	$M_{p}=15\%$	$S_{f}=unchanged$	$N_{p}=10\%$	$P_{d}=0$	$O_{p}=0\%$
Scenario V	$M_{p}=15\%$	$S_{f}=unchanged$	$N_{p}=0\%$	$P_{d}=10$ m/min	$O_{p}=0\%$
Scenario VI	$M_{p}=15\%$	$S_{f}=unchanged$	$N_{p}=0\%$	$P_{d}=0$	$O_{p}=10\%$

## Data Availability

The data supporting this study is available upon request from the corresponding author.

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
