# Peer review of "GTR: GAN-Based Trusted Routing Algorithm for Underwater Wireless Sensor Networks"

_sensors, 2024, doi:10.3390/s24154879_

Round 1

Reviewer 1 Report

Comments and Suggestions for Authors

See the attached file for comment.

Reviewer 2 Report

Comments and Suggestions for Authors

This article presents a GAN-based Trusted Routing Algorithm for Underwater 2 Wireless Sensor Networks. However, I have some comments as follows:

1.      The abstract is succinct and coherent, offering a comprehensive summary of the issue, suggested resolution, and outcomes. However, it would be advantageous to explicitly state the primary innovation of the proposed algorithm in comparison to existing methodologies.

2.      It is suggested to add supplementary diagrams or flowcharts that could augment the lucidity of the algorithmic procedures and the overall framework.

3.      The figures and tables are meticulously arranged and provide strong support to the narrative. But certain visual representations could be enhanced by increasing the resolution and providing more elaborate captions to guarantee their complete comprehensibility. Moreover, the resolution of all the figures must be improved, especially 10-14.

4.      In the introduction section, it would benefit from incorporating more up-to-date citations to emphasize the present status of research in this domain. The authors can also refer to "a review of underwater localization techniques, algorithms, and challenges" and "localization and detection of targets in underwater wireless sensor using distance and angle based algorithms".

5.      The literature review would be improved by doing a thorough examination of the constraints of current approaches, which would more effectively validate the necessity of the suggested methodology.

6.      Further elaboration is needed for the incorporation of Generative Adversarial Networks (GAN) with Q-Learning on how the GAN model precisely tackles the issues in UWSNs, notably the reasoning behind choosing GAN over other machine learning models.

7.      It would be advantageous to include further information regarding the simulation environment and any underlying assumptions established during the studies.

8.      The results are displayed lucidly, demonstrating the performance increases achieved by the GTR algorithm. However, the analysis might be enhanced by offering a more comprehensive comparison with baseline algorithms, which should include statistical significance checks to justify the observed gains.

9.      The discussion section should go more into possible constraints of the study and propose avenues for future research.

10.   The conclusion section could additionally underscore the pragmatic implementations of the suggested method and its potential ramifications on real-world UWSNs.

11.   Finally, the overall article should be revised for detailed grammatical and typo errors.

Comments on the Quality of English Language

English must be improved

Round 2

Reviewer 2 Report

Comments and Suggestions for Authors

The authors have already addressed all the comments. I have no further comments. 

Comments on the Quality of English Language

Moderate editing of English language required